# Recent Advances in the Hydroxylation of Amino Acids and Its Derivatives

**Bangxu Wang** [1,†], **Shujian Xiao** [1,†], **Xingtao Zhao** [1], **Liming Zhao** [2], **Yin Zhang** [1], **Jie Cheng** [1,*] and **Jiamin Zhang** [1,*]

1   Meat Processing Key Laboratory of Sichuan Province, College of Food and Biological Engineering, Chengdu University, Chengdu 610106, China
2   State Key Laboratory of Bioreactor Engineering, East China University of Science and Technology, Shanghai 200237, China
*   Correspondence: chengjie@cdu.edu.cn (J.C.); zhangjiamin@cdu.edu.cn (J.Z.)
†   These authors contributed equally to this work.

**Abstract:** Hydroxy amino acids (HAAs) are of unique value in the chemical and pharmaceutical industry with antiviral, antifungal, antibacterial, and anticancer properties. At present, the hydroxylated amino acids most studied are tryptophan, lysine, aspartic acid, leucine, proline, etc., and some of their derivatives. The hydroxylation of amino acids is inextricably linked to the catalysis of various biological enzymes, such as tryptophan hydroxylase, L-pipecolic acid trans-4-hydroxylase, lysine hydroxylase, etc. Hydroxylase conspicuously increases the variety of amino acid derivatives. For the manufacture of HAAs, the high regioselectivity biocatalytic synthesis approach is favored over chemical synthesis. Nowadays, the widely used method is to transcribe the hydroxylation pathway of various amino acids, including various catalytic enzymes, into *Corynebacterium glutamicum* or *Escherichia coli* for heterologous expression and then produce hydroxyamino acids. In this paper, we systematically reviewed the biosynthetic hydroxylation of aliphatic, heterocyclic, and aromatic amino acids and introduced the basic research and application of HAAs.

**Keywords:** biosynthesis; hydroxylation; hydroxylated amino acid; 5-Hydroxytryptamine; hydroxylysine





## 1. Introduction

An amino acid (AA) is a cell signal molecule with significant metabolic and regulatory functions [1]. They are necessary precursors for the synthesis of various important molecules. They can be converted into various derivatives through halogenation, *N*-alkylation, hydroxylation, and other ways. These derivatives have important roles in various fields, such as medicine and the chemical industry [2]. Among them, hydroxy amino acids play an important role. Hydroxylation of amino acids is one of the most common C-H bond functionalization reactions. However, the general inertness of the C-H bond makes it one of the most challenging reactions to selectively convert it into highly regioselectivity hydroxy amino acids (HAAs) [3,4]. In this context, the development of engineering enzymes to control the selective synthesis of HAAs has become an attractive method for amino acid hydroxylation. For example, employing the Fe(II)/α-ketoglutarate-dependent dioxygenases (KDOs) could fine-catalyze the generation of hydroxylysine [5]. In addition, Arginine hydroxylase (VioC) catalyzes the generation of 3-hydroxy-arginine (3-OH-Arg) through protein engineering and recombinant gene engineering [6].

HAAs have recently become widely used in various industries, including biology, medicine, food, cosmetics, and others (Figure 1). For example, hydroxyl-lysine, *trans*-4-hydroxyl-proline, and hydroxyl-isoleucine are good pharmaceutical raw materials [7]. L-5-hydroxytryptophan (5-HTP) increases melatonin levels and promotes sleep [8]. 3-Hydroxy-L-tyrosine can be used as a drug for tremors, and hydroxyproline can be used to prepare cosmetics [9,10]. The research of screening, extracting, and purifying hydroxylase

for producing HAAs is the current research hotspot. This work reviews the latest research progress of the hydroxylated synthesis of aromatic amino acids, aliphatic amino acids, and heterocyclic amino acids in Table 1. In addition, the applications of various HAAs are introduced and prospected in detail.

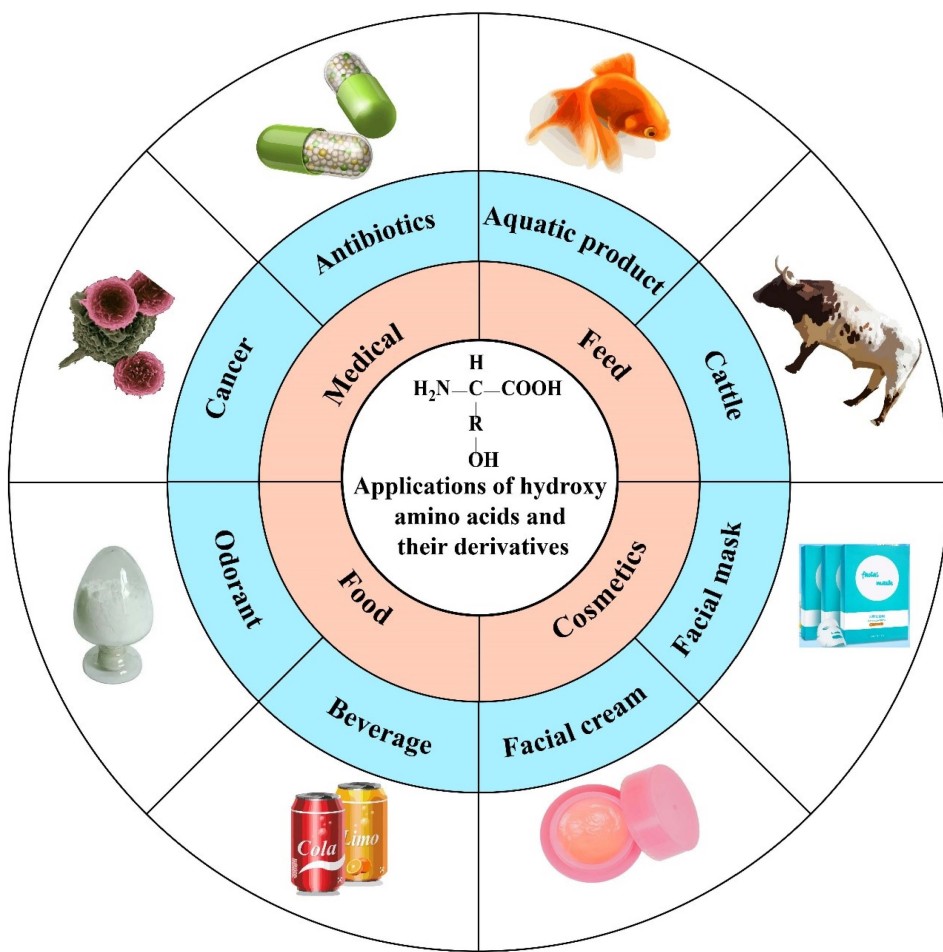

**Figure 1.** The application of hydroxy amino acids and their derivatives.

**Table 1.** The enzymatic hydroxylation of amino acids and their derivatives.

| Product | Chemical Structure | Host | Substrate | Titer (g/L) | Strategy | Reference |
|---|---|---|---|---|---|---|
| 3,4-Dihydroxyphenyl-L-alanine | | *Corynebacterium glutamicum* | L-Tyrosine | 0.26 | Heterologous expression of tyrosinase from *Ralstonia solanacearum* | [10] |
| 5-HTP | | *E. coli* | L-Serine | 17.79 | Directed evolution of tryptophan synthase | [11] |
| 5-HTP | | *E. coli* | L-Tryptophan | 5.1 | Expression of TPH1 | [12] |

**Table 1.** *Cont.*

| Product | Chemical Structure | Host | Substrate | Titer (g/L) | Strategy | Reference |
|---|---|---|---|---|---|---|
| Hydroxytyrosol | | *E. coli* | Glucose | 4.69 | Directed Optimization of VanR and TYO | [13] |
| Tyrosol | | *E. coli* | Tyrosine | 1.18 | Introducing a yeast pyruvate decarboxylase ARO10 into *E. coli* | [14] |
| (3*S*)-3-Hydroxy-L-lysine | | *E. coli* | L-Lysine | 5 | Screening and identifying novel KDOs | [15] |
| (2*S*,3*S*)-3-Hydroxylysine | | *E. coli* | L-Lysine | 86.1 | Gene mining and phylogenetic analysis from Clavaminic acid synthase-like superfamily | [16] |
| (2*S*,4*R*)-4-Hydroxylysine | | *E. coli* | L-Lysine | 43.0 | Gene mining and phylogenetic analysis from Clavaminic acid synthase-like superfamily | [16] |
| *cis*-3HPA | | *E. coli* | L-Lysine | 47.9 | Expression of GetF and SpLCD | [5] |
| *cis*-5HPA | | *E. coli* | L-Lysine | 13.95 | Established a complete biological method | [17] |
| 3-OH-Arg | | *E. coli* | L-Arginine | 9.9 | Using protein engineering and recombinant whole-cell biocatalysis | [18] |
| 4-HIL | | *E. coli* | L-Isoleucine | 10.83 | Cloning of the IDO gene from *Bacillus thuringiensis* and *B. thuringiensis* strains | [19] |
| 5-HLeu | | *E. coli* | L-Leucine | 18.83 | Constructing two bioconversion systems | [20] |
| c3Hyp | | *E. coli* | L-Proline | 9.1 | Cloning and heterologous expression of P3H | [21] |
| *Trans*-4-hydroxy-L-proline | | *Corynebacterium glutamicum* | Glucose | 7.1 | Constructed a plasmid (pEKE_p4h1of) containing the P4H gene | [22] |
| *Trans*-4-hydroxy-L-proline | | *Corynebacterium glutamicum* | L-proline | 0.113 | Cloned and expressed the genes of trans-proline 4-hydroxylase from diverse resources | [7] |
| 5-Hydroxylysine | | *Corynebacterium glutamicum* | Lysine | 0.88 | Overexpressed KDOs genes | [23] |

## 2. Hydroxylation of Aromatic Amino Acids

### 2.1. Hydroxylation of Tryptophan and Tryptophan Derivatives

5-HTP is a specific amino acid found in animals and plants. 5-HTP could increase the content of brain serum and melatonin and promote sleep [8]. 5-HTP could cross the blood-brain barrier and reduce the level of cytokine in the body, thus improving depression [24,25]. Xu et al. isolated and purified the tryptophan synthase from *E. coli* by a direct evolutionary strategy [11]. Tryptophan synthase catalyzes L-Serine and 5-hydroxy-indole to synthesize 5-HTP. The yield of 5-HTP was 86.7% under the conditions of 100 mM L-serine, 35 °C for 15 h [11]. Interestingly, Vargas et al. found that adding surfactant Tween 20 in the culture broth can expand the pH operation range of tryptophan hydroxylase for producing 5-HTP [26]. 6-Hydroxy-L-tryptophan, an inhibitor of tyrosinase in vitro, has not been found in natural resources.

5-hydroxytryptamine (5-HT) is one of the tryptophan derivatives, also known as serotonin, which is synthesized in the central nervous system and the serotonin neurons of the gastrointestinal tract [12,27]. 5-HT is crucial for regulating anger, appetite, aggression, body temperature, mood, sexual behavior, and sleep [27]. Reigstad et al. studied the potential mechanism of microbial products promoting Tph1 and 5-HT synthesis using a human EC cell model [27]. Wang et al. first expressed the human tryptophan hydroxylase I (TPH1) and reconstructed the regeneration pathway of tetrahydrobiopterin. Finally, 1.2 g/L of 5-HT was produced by whole-cell biotransformation from 2 g/L L-tryptophan [12].

### 2.2. Hydroxylation of Phenylalanine and Phenylalanine Derivatives

L-3-hydroxy-phenylalanine is associated with Parkinson's disease, Alzheimer's disease, and arthritis [28]. The naturally occurring L-2-hydroxy-phenylalanine and L-3-hydroxy-phenylalanine are rare and could be generated by the non-enzymatic free radical hydroxylation of phenylalanine under oxidative stress [29]. Of course, enzymatic catalysis has also been reported for the synthesis of L-3-hydroxy-phenylalanine. For example, the $Fe^{2+}$-dependent phenylalanine hydroxylase (PheH) can catalyze the hydroxylation of aromatic amino acid L-phenylalanine to L-3-hydroxy-phenylalanine [30].

Hydroxytyrosol, 3,4-dihydroxy-phenylethanol, is a phenolic alcohol thought to exert a wide range of biological effects, including cardioprotective, anticancer, neuroprotective, antimicrobial, and endocrine effects. The precise molecular mechanisms behind many of these impacts have not yet been fully clarified, despite the extensive investigations that have been done. Initially, diverse hydroxytyrosol bioactivities were associated with its potent antioxidant activity. Hydroxytyrosol acts as a free radical scavenger and metal chelating agent. The presence of O-dihydroxyphenyl molecules attributes to the high antioxidant efficacy of hydroxytyrosol. Yao et al. designed an efficient whole-cell catalyst for producing hydroxytyrosol in *E. coli*, eliminating the bottleneck of two rate-limiting enzymatic steps [13]. First, they used structure-guided modeling and directed evolution to replace the mouse tyrosine hydroxylase with the two-component flavin-dependent monooxygenase HpaBC from *E. coli.* The next step was to elucidate the structure of the complex between the regulatory protein VanR of glutamic acid and its inducer, vanillic acid. By converting its inducible specificity from vanillic acid to hydroxytyrosol, VanR was developed as a biosensor for detecting hydroxytyrosol. By using this biosensor to control evolution in vivo, it was possible to increase the activity of tyramine oxidase (TYO), the second rate-limiting enzyme for hydroxytyrosol production, to 95%, in the final strain.

Tyrosol, 2-(4-hydroxyphenyl)-ethanol is a potent cellular antioxidant. Tyrosol is less subject to auto-oxidation than other polyphenols because it is a relatively stable compound. Even under critical conditions, it maintains its antioxidant activity. Tyrosol retains its antioxidant effect after autoxidation has already begun in the presence of oxidized LDL. However, other more potent natural flavonoids see a significant decline in antioxidant activity, occasionally even transitioning to pro-oxidants. Since the extraction of tyrosol from plants poses a major obstacle, Xue et al. obtained two recombinant tyrosol-producing strains by introducing the phenylpyruvate decarboxylase gene *ARO10* and the aromatic

amino acid transaminase gene *ARO8* into *E. coli* [14]. The prephenate dehydratase gene *pheA* and the phenylacetaldehyde dehydrogenase gene *feaB* were deleted to enhance the titer of tyrosol. The recombinant strain overexpressing the *ARO10* gene in a shake flask incubated with 1% (*w/v*) glucose for 48 h produced 4.15 mM tyrosol. The recombinant strain co-expressing the *ARO8* and *ARO10* genes had a higher tyrosol titer when tyrosine was used as a substrate, and 8.71 mM tyrosol was obtained from 10 mM tyrosine.

### 2.3. Hydroxylation of Tyrosine and Tyrosine Derivatives

3-Hydroxy-L-tyrosine (L-DOPA) is one of the precursors for the synthesis of norepinephrine and dopamine in the body. It belongs to catecholamine, which can be used as a drug to treat tremors. L-DOPA crosses the blood-brain barrier into the brain and is converted to dopamine by decarboxylation with equine decarboxylase. Kurpejović et al. reported that L-DOPA could be produced by heterologous expression of *Ralstonia solanacearum* tyrosinase in *Corynebacterium glutamicum* from glucose or glucose/xylose mixtures [10].

Dopamine (3-hydroxytyramine) is the most abundant catecholamine neurotransmitter in the brain and acts as a neurotransmitter regulating a variety of physiological functions of the central nervous system. Disorders of dopamine system regulation are involved in Parkinson's disease, Tourette's syndrome, schizophrenia, attention deficit hyperactivity syndrome, and the development of pituitary tumors. Dopamine is a neurotransmitter in the brain related to human lust and sensation, and it transmits excitement and happy messages.

## 3. Hydroxylation of Aliphatic Amino Acids and Their Derivatives

### 3.1. Hydroxylation of Lysine

Hydroxylysine (Hyl), a product of collagen degradation, is a building block for the synthesis of a variety of pharmacologically relevant molecules and also be used as a raw material for collagen synthesis, which is critical to the safety and effectiveness of many pharmaceutical products. As seen in Figure 2, the synthesis of Hyl is generally based on lysine as raw material, and the oxidation of the C-H bond to the C-OH bond through hydroxylation reaction catalyzed by hydroxylase, which can introduce hydroxyl groups at $C_3$, $C_4$, and $C_5$ positions to form hydroxylysine. Rolf et al. screened and characterized a new KDO for the whole-cell conversion synthesis of (3S)-3-hydroxy-L-lysine [15]. Two L-lysine-3S-hydroxylases and four L-lysine-4R-hydroxylases were analyzed and identified by mining and phylogeny for lysine $C_3$, $C_4$ hydroxylation reactions from the clathrate synthase-like superfamily genes. These hydroxylases can be used for industrial bulk synthesis of hydroxylysine at reduced synthesis cost [16]. Interestingly, Seide et al. proposed a covalent in situ immobilized strategy to form (3S)-hydroxy-lysine and (2S)-hydroxy cadaverine by the immobilization of dioxygenase and lysine decarboxylase [31]. Two lysine dioxygenases, KDO1 and KDO5, with different regioselectivity in the Clavaminate Synthase-Like Family, were reported to catalyze the formation of (3S)-3-hydroxy-L-lysine and (4R)-4-hydroxy-L-lysine, respectively [32].

(2S,4R)-4-Hydroxylysine is a highly versatile intermediate used to synthesize many important pharmaceutical compounds. Wang et al. modified the t-lysine hydroxylase of *Niastella koreensis* (NkLH4) by CAST to obtain the highly active mutant MT3, which has a more flexible conformation that expands the substrate binding pocket of the hydroxylation reaction, reduces steric hindrance, and increases the binding energy in substrate recognition [33]. It is used to catalyze (2S,4R)-4-hydroxylysine and increase the reaction rate. Prell et al. reported that the biolase KDO from *Flavobacterium johnsoniae* catalyzes the reaction to generate 4-hydroxylysine [23]. A lysine 4-hydroxylase (GlbB) was synthesized in the gene cluster of griddopa actin biosynthesis, capable of catalyzing the hydroxylation of L-lysine with an excellent total turnover number and complete regioselectivity and diastereoselectivity [34]. Baud et al. preselected and screened for the isolation of five new KDOs using a genomic approach. Four KDOs were active against L-lysine and produced the corresponding 3-HAAs or 4-HAAs, and the enzymatic cascade reaction with two stereoselective KDOs allowed the synthesis of 3,4-dihydroxy-L-lysine [35].

**Figure 2.** The hydroxylation of lysine and its derivatives. KDO, Fe(II)/α-ketoglu tarate-dependent dioxygenases; α-KG, α-ketoglutarate; KDO$_{Fj}$, α-ketoglutarate dependent dioxygenase from *Flavobacterium johnsoniae*; KDO$_{Ca}$, α-ketoglutarate dependent dioxygenase from *Catenulispora acidiphila*; KDO$_{Kr}$, α-ketoglutarate dependent dioxygenase from *Kineococcus radiotolerans*; KDO$_{Nk}$, α-ketoglutarate dependent dioxygenase from *Niastella koreensis*; KDO$_{Cp}$, α-ketoglutarate dependent dioxygenase from *Chitinophaga pinensis*; KDO$_{Cg}$, α-ketoglutarate dependent dioxygenase from *Chryseobacterium gleum*; KDO$_{Pl}$, α-ketoglutarate dependent dioxygenase from *Photorhabdus luminescens*; VioC, L-arginine 3-hydroxylase; PiFa, L-pipecolic acid hydrolylase from *Frankia alni*; RaiP, L-lysine a-oxidase; DpkA, Δ$^1$-piperideine-2-carboxylase reductase; SmP4H, *cis*-4-hydroxylases from *Sinorhizobium meliloti*; FoPip4H, L-pipecolic acid *trans*-4-hydroxylase; GetF, L-pipecolic acid hydroxylase; 2K6AC, 2-keto-6-aminocaproate; Pip2C, Δ$^1$-piperideine-2-carboxylic acid; L-Pip, L-pipecolic acid; *cis*-3HPA, *cis*-3-hydroxypipecolic acid; *Trans*-4HPA, *trans*-4-hydroxy-L-pipecolic acid; *cis*-5HPA, *cis*-5-hydroxypipecolic acid.

5-hydroxylysine (5-Hyl) is an important pharmaceutical intermediate with a variety of regional and stereoisomers. Allevi et al. controlled the four-step synthesis of (2S,5R)-5-Hyl and (2S,5S)-5-Hyl by using Williams' glycine template approach for stereoisomerism at the a-position [36]. This route offers the possibility of synthesizing 5-Hyl and all possible isomers of the important α-amino acid iodohydrin required for the construction of the hydroxylated side chain of the collagen cross-linked pyridinoline [36].

### 3.2. Hydroxylation of Lysine Derivatives

The hydroxylated products of lysine derivatives are widely used in pharmaceuticals and chemicals. Pipecolic acid is one of these lysine derivatives, and after being hydroxylated, it can be utilized as a marker for neurological disorders as well as an intermediate in the production of various medications and antibiotics. Hydroxypipecolic acid (HPA) is a crucial component of organic synthesis and metabolite present in plants and animals. Many experts and scholars are working to improve the status of HPA production by studying HPA biosynthetic enzymes, etc. Hu et al. developed and utilized lysine cyclic deaminase (SpLCD) to cyclically deaminate L-lysine to L-pipecolic acid, which was further hydroxylated by KDO (GetF) for whole-cell conversion to *cis*-3-hydroxypipecolic acid (*cis*-3HPA), reducing the production cost of HPA [5]. Cheng et al. established an artificial synthesis of *trans*-4-hydroxy-L-pipecolic acid (*Trans*-4-HPA) by overexpressing lysine α-oxidase (DpkA), Δ-1-pipecolic-2-carboxylase reductase (LysP), L-pipecolic acid *trans*-4-hydroxylase (FoPip4H) and other enzymes in *E. coli* [37]. Filamentous fungi were discovered to contain

a new family of enzymes that catalyze the trans-4-hydroxylation of L-pipecolic acid. These enzymes were cultivated and examined to learn how to make trans-4-L-HPA [38].

An essential chiral step for producing the β-lactamase inhibitor avibactam is cis-5-hydroxy-L-pipecolic acid (cis-5HPA) (Avibactam). Moreover, it can be employed as a primitive substrate to make the production of avibactam much simpler. Lu et al. optimized the hydroxylation process of L-pipecolic acid into *cis*-5-HPA by catalyzing the hydroxylation of L-pipecolic acid with KDOs and other enzymes in the whole cell in water, thus reducing the oxidation of $Fe^{2+}$, optimizing the biological transformation conditions, and improving the catalytic efficiency [17]. After two steps of eliminating Fe(II) ions and reaction-related impurities and isomer separation, pure cis-5-HPA was obtained [17].

### 3.3. Hydroxylation of Arginine

3-OH-Arg, a central intermediate, could be used to synthesize the antibiotic viomycin. The subsequently generated antibiotic viomycin can be used in the clinical treatment of tuberculosis. An arginine hydroxylase from *Streptomyces vinaceus* (VioC), which can catalyze the synthesis of 3-OH-Arg from L-arginine (L-Arg), was found to be expressed in *E. coli* BL21(DE3) during a study of gene clusters related to viomycin biosynthesis [39–41]. Mao et al. proposed an efficient 3-OH-Arg production strategy based on protein engineering and recombinant whole-cell biocatalysis by designing a pathway for synthesizing 3-OH-Arg catalyzed by the recombinant strains LGOX, CAT, and ODO using L-glutamate (L-Glu) and L-Arg as substrates. In detail, Mao et al. optimized the molar ratio of the substrates L-Arg and L-Glu to ensure the effective production of 3-OH-Arg and the thorough consumption of α-KG. This strategy prevents the challenge of product isolation caused by the creation of α-KG in the system [18]. Baud et al. preselected and screened for the isolation of five new KDOs using a genomic approach. Four KDOs of them KDOs were active against L-Arg and produced the corresponding 3-OH-Arg [35]. Yin et al. heterologously expressed a non-heme iron VioC in *E. coli*, in Viomycin biosynthesis utilizes free l-arginine as a substrate and generates 3-OH-Arg as a product [6].

### 3.4. Hydroxylation of Aspartic Acid

Hydroxyaspartic acid (HO-Asp) is used for synthesizing drugs and antibiotics. 3-hydroxyaspartic acid and its derivatives exist in free form and peptide components in various microorganisms and fungi. This amino acid is of great biological importance and has the potential to serve as a multifunctional building block in organic synthesis. Liu et al. reported an efficient and practical synthetic route to obtain a variety of protected erythro-OH-Asp, which are key β-branched α-amino acid units in coralloidin A and other peptide natural products. The Fmoc- and cyclic ketone-protected erythro-OH-Asp were produced in six steps without column purification using the inexpensive chiral precursor diethyl L-tartrate as a starting material. They used diethyl tartrate as a starting material because of its low cost as an easily available chiral precursor [42].

Hara et al. effectively synthesized L-threo-3-hydroxyaspartic acid using microbial hydroxylase and hydrolase (L-THA) [43]. Although they have a relatively low yield, asparagine hydroxylase (AsnO-D241N) and its homolog (SCO2693-D246N) show good direct hydroxylation characteristics for L-aspartic acid. Therefore, they used the highly stable and effective wild-type asparagine hydroxylase AsnO and SCO2693 to synthesize L-THA. The recombinant enzyme hydroxylated L-asparagine by AsnO and then hydrolyzed by 3-hydroxy-asparagine amide to obtain L-THA. Subsequently, one-pot biotransformation was performed in a test tube using a two-step reaction. Tiny amounts of L-THA, with a molar yield of 0.076%, were generated in two asparaginase-defective mutants expressing the AsnO gene. In the asparaginase i-deficient mutant, L-THA production was significantly increased by 8.2%. L-THA production was increased to 92% in *E. coli* by using the T7 promoter instead of the lac promoter to enhance the expression level of the AsnO gene. Using a combination of the *E. coli* asparaginase i-deficient mutant and the T7 expression system, a whole-cell reaction was performed in a tank fermenter, culminating in the

successful production of L-THA from L-asparagine in yields of up to 96% in less time than test-tube scale production. These results suggest that asparagine hydroxylation hydrolysis can be used for the efficient production of L-THA [43].

### 3.5. Hydroxylation of Aspartic Acid Derivatives

Derivatives of HO-Asp are usually reported as byproducts of the reaction. Chen et al. found the ambiguous structures of methoxyaspartic acid (MeO-Asp) and hydroxyaspartic acid (HO-Asn) reported in the total synthesis of A54145 [44]. Chen et al. have developed efficient routes to provide fully protected L-MeO-Asp and L-HO-Asn building blocks compatible with Fmoc-SPPS and the total synthesis of A54145. In addition, their results allow us to determine its structure, consisting of L-3*S*-HO-Asn and L-3*R*-MeO-Asp, and revise the erroneously proposed structure of L-3*S*-HO-Asn. L-HO-Asn is a non-protein-derived amino acid found in many natural products. L-HO-Asn building blocks with various protecting groups have been chemically synthesized using a number of enantioselective synthetic techniques. However, the fully protected L-HO-Asn used for Fmoc-SPPS has been protected with Trt (tributyl) and TRS (tert-butyldimethylsilyl) to protect the amide and hydroxyl groups of the side chain, respectively.

Jungmann et al. demonstrated a high level of hydroxyoctanoic acid production using a metabolically engineered strain of glutamate expressing a codon-optimized heterologous ectD gene encoding an octanoic acid hydroxylase that converts supplemented octanoic acid into the desired derivative as a growth substrate in the presence of sucrose. Fourteen of 16 codon-optimized ectD variants from different phylogenetic bacterial and archaeal donors achieved hydroxylimine production, indicating that the strategy is virtually independent of gene origin. Genes from *Pseudomonas aeruginosa* (PST) and *Mycobacterium smegmatis* (MSM) worked best, resulting in 97% hydroxyeicosanoid production. Metabolic analysis revealed a high enrichment of intracellular isoflavones, which, among others, reduced the synthesis of other compatible solutes, including proline and alginate. After further optimization, glutamate Ptuf ectDPST reached 74 g/L of hydroxyethyl octane at 70% selectivity within 12 h using a simple batch procedure. Hydroxyoctane was produced in a two-step procedure from ectoine previously synthesized by fermentation with *C. glutamicum* ectABC. *C. glutamicum* ectABCopt, the production of hydroxyethyl octane has been successfully achieved without the need for an intermediate purification step [45].

### 3.6. Hydroxylation of Glutamic Acid

Hydroxyglutamate mainly exists in specific proteins as an antibiotic precursor. Matthias et al. cloned and recombinant expressed two putative nonheme iron oxygenases, KtzO and KtzP, in *E. coli* and biochemically characterized them in vitro and found that they stereospecifically hydroxylated the β position of glutamate, and as a result, generated Threo- and erythro-β-OH-glutamate acids [46].

### 3.7. Hydroxylation of Leucine and Isoleucine

Hydroxyleucine (HLeu) is usually a substructure and biosynthetic precursor of certain bioactive natural products, which can be widely used in food, chemical, and other fields. Hydroxylase enzymes catalyze the hydroxylation of leucine and isoleucine to introduce hydroxyl groups in hydroxyisoleucine (HIL) and HLeu through deoxygenated C-H bonds. Ries et al. developed a method for the synthesis of (2S,3S)-3-hydroxyleucine building blocks as part of work on the total synthesis of wall mycin nucleoside antibiotics [47]. Sun et al. investigated the use of L-leucine dioxygenase (LDO, NpLD0), L-glutamate oxidase (LGOX), and catalase (CAT) to synthesize a synthetic route for the simultaneous production of SA and 5-hydroxyleucine (5-HLeu) using L-Leu. Two biotransformation systems were constructed, namely in vitro multi-enzyme cascade catalysis (MECCS) and whole cell catalysis (WCCS) [20]. Smirnov et al. synthesized 4-HLeu and 4-HIL by expressing cloned L-isoleucine-4-hydroxylase (IDO) for L-leucine and L-isoleucine in *E. coli*. It was shown that a new subfamily of bacterial dioxygenases exists in the PF10014 family, in which free L-amino

acids can be used as substrates in vivo. The physiological significance of hydroxylated L-amino acids has also been discussed [48]. Hibi et al. established a new enzymatic production system for β-hydroxy-α-amino acids by stereoselective hydroxylation of several *N*-succinyl aliphatic L-amino acids using *N*-succinyl L-amino acid hydroxylase (SadA), SadA stereoselectively hydroxylated several *N*-succinyl aliphatic L-amino acids such as *N*-succinyl-L-hydroxyvaline and produced N-succinyl β-hydroxy L-amino acids, such as *N*-succinyl-L-β-hydroxyvaline, (2*S*,3*R*)-*N*-succinyl-L-β-hydroxyisoleucine, and *N*-succinyl-L-threo-β-hydroxyleucine [49]. 4-hydroxyisoleucine (4-HIL) is a naturally occurring non-protein-derived amino acid with insulinogenic biological activity. Smirnov et al. cloned and expressed stereospecific IDO in an *E. coli* 2Δ strain lacking the activities of α-KG dehydrogenase, the activity of IDO from 2Δ strain can "shunt" the disrupted TCA, thus converting L-isoleucine hydroxylation to 4-HIL [19].

*3.8. Hydroxylation of Leucine and Isoleucine Derivatives*

Derivatives of HLeu and HIL are commonly applied as drugs and antibiotics. For instance, the (2*S*,3*S*)-3-hydroxyleucine derivatives are designed for further modification in the carboxyl and amino moieties and for solid-phase peptide synthesis for the formation of antibiotics and drugs, etc. Ries et al. generated building blocks suitable for *C*- or *N*-terminal derivation as well as solid-phase peptide synthesis by applying different protecting group patterns [47]. The building blocks were transformed into 3-*O*-acylated structures for the corresponding motifs present in the natural products. Using esterification and cross-recombination protocols, (2S,3S)-3-Hydroxyleucine derivatives have been synthesized. This provides an excellent route to synthesizing biologically active natural products and their derivatives for Structure-Activity Relationship (SAR) studies [47]. Hansen et al. synthesized (2*S*,4*R*)-δ-hydroxyleucine and (2*S*,4*R*)-δ-hydroxyleucine methyl ester by asymmetric alkylation with EvansO and asymmetric Strecker reaction with Daviso, and these could be used as raw materials to produce cytotoxic cyclomatic A against cancer cells [50]. Wu et al. resolved that IDO directly catalyzes the C-H bond hydroxylation of several hydrophobic aliphatic amino acids. However, its application in the production of chiral hydroxylated amino acids is hampered by the poorly defined selectivity of IDO. The hydroxylation of L-norleucine by IDO to produce 4-hydroxynorleucine and 5-hydroxynorleucine with distinct regioselectivity was used to investigate the mechanism of IDO regioselectivity. IDO structures revealed single site variants (T244A, T244G, and T244S) with enhanced regioselectivity through computational structure analysis and high throughput screening. For example, the purity of 4-hydroxynorleucine in the regiospecific products increased from 78.9% (wild-type IDO) to 95.1%, 96.6%, and 95.3%, respectively. Molecular dynamics simulations suggest that the mutation of T244 to smaller amino acids allows fine-tuning of the substrate binding position. This change increased the most common distance between substrate $C_4$ or $C_5$ and $Fe^{2+}$, resulting in a maximum purity of 96.6% for 4-hydroxynorleucine for asymmetric catalysis requiring precise positioning. This study improves the understanding of the regioselectivity of KDOs and provides a way to diversify the C-H hydroxyl group active compounds [51]. Zwick et al. found that a leucine derivative, manzacidin C, was enzymatically synthesized by a promiscuous leucine 5-hydroxylase from *Streptomyces muensis* for providing an application in complex molecule synthesis [52].

## 4. Hydroxylation of Heterocyclic Amino Acids and Their Derivatives

*Hydroxylation of Proline*

Hydroxyprolines (Hyps) are one of the main components of collagen tissue and contain six isomers, including *trans*-4-hydroxy-proline (t4Hyp) [22], *trans*-3-hydroxy-L-proline (t3Hyp), *cis*-4-hydroxy-L-proline (c4Hyp), *cis*-3-hydroxy-L-proline (c3Hyp), *trans*-5-hydroxy-L-proline (t5Hyp), and *cis*-5-hydroxy-L-proline (c5Hyp). Hyps are unique amino acids in collagen, which are important components of animal collagen. It is also found

in many plant proteins and is particularly associated with cell wall formation. Hyps are widely used in the pharmaceutical, nutritional, and cosmetic industries [9].

The hydroxylation of proline is catalyzed by a family of KDOs, which catalyze its hydroxylation reaction by biological enzymes. This is because the central carbon atom of hydroxylation in proline is prone to the formation of stereoisomerism during the hydroxylation process. Its hydroxylation has also been studied, centering on the hydroxylation reactions of different isomers. Zhang et al. performed biochemical characterization of KDO (HtyE) in two *Aspergillus species* (*Aspergillus pachycristatus*) and Aculeatus in echinocandins. The results showed that both Ap-HtyE and Aa-HtyE converted L-proline to t4Hyp and t3Hyp, and both enzymes also efficiently hydroxylated (4*R*)-methyl-proline, L-pipeline acid, and D-proline. Their study revealed the biochemical basis of selective hydroxylation of L-proline and (4*R*)-methyl-proline and revealed a novel biocatalyst. This catalyst could be used in the future for the tailor-made production of hydroxylated derivatives of proline and pipecolic acid [53]. Klein et al. established an in vivo program in *Escherichia coli* to prepare c3Hyp, c4Hyp, and t4Hyp hydroxylases for hydroxylation of L-proline by cloning, protein expression, transformation, analysis, and product purification [54]. Davies et al. developed two routes for the asymmetric synthesis of (2*R*,3*S*)-3-hydroxyproline and (2*S*,3*S*)-3-hydroxyproline by converting protected α,δ-dihydroxy-β-amino esters (2,3-*trans* configurations or 2,3-*cis* configurations) to β,δ-dihydroxy-α-amino esters through the mediation of the corresponding aziridinium ions [55]. The products of these stereospecific rearrangements were then cyclized and deprotected. The yield of 3-hydroxyproline in different configurations was further improved to provide ideas for the synthesis of similar organic compounds [55]. Dioxidase-proline-4-hydroxylase (P4H) is over-synthesized by recombinant *E. coli* for the synthesis of t4Hyp in large quantities in whole cells [56]. Johnston et al. constructed a synthetic gene encoding Streptomyces L-proline-3-hydroxylase by biocatalytic synthesis for the production of hydroxylase protein in recombinant *E. coli* and for the synthesis of c3Hyp [21].

Pérez-Fernández et al. used serine cyclobutane derivatives as raw materials and synthesized 5-hydroxyproline by Michael addition and Wittig reaction [57]. Zhang et al. developed an efficient total proline-proline dehydrogenase cascade for the enantioselective production of D-proline. Efficient dehydrogenation of DL-proline was achieved using whole cells of Pseudomonas aeruginosa XW-40 with proline dehydrogenase, and *N*-boc-5-hydroxy-L-proline was synthesized from the l-proline dehydrogenation product L-Glu semialdehyde. The proline racemase-proline dehydrogenase cascade developed in the study has great potential and economic competitiveness for the synthetic manufacture of *N*-boc-5-hydroxy-L-proline [58].

## 5. Conclusions

HAAs are an important class of functional amino acids with a variety of functions and are widely used in the feed, pharmaceutical, building, and chemical industries. It not only has antifungal, antibacterial, antiviral, and anticancer effects but also can be used as a precursor or intermediate in the asymmetric synthesis of drugs and can also be used as a component of chiral compounds. Based on the widespread use of HAAs, the development of engineered enzymes to control the selective synthesis of HAAs is one of the indispensable methods to promote the yield of HAAs. Many engineered enzymes have been found to hydroxylate free amino acids into corresponding HAAs. However, the biocatalytic synthesis of HAAs is still limited to a single enzyme and a single carrier. Therefore, new members of functional amino acid hydroxylases should be characterized by methods involving gene mining, bioinformatics, and high-throughput screening techniques.

**Author Contributions:** Writing—original draft preparation, B.W., S.X. and X.Z.; writing—review and editing, Y.Z. and J.C.; visualization, J.Z.; supervision, J.Z.; project administration, J.Z. and J.C.; funding acquisition, L.Z. and J.C. All authors have read and agreed to the published version of the manuscript.

**Funding:** This work was supported by the National Natural Science Foundation of China (22108017), the Natural Science Foundation of Sichuan Province (2022NSFSC1614), the Open Funding Project of the State Key Laboratory of Bioreactor Engineering, and Partially Supported by the Chengdu University Undergraduate Innovation Training Plan Incubation and Cultivation Project.

**Institutional Review Board Statement:** Not applicable.

**Informed Consent Statement:** Not applicable.

**Data Availability Statement:** Not applicable.

**Conflicts of Interest:** The authors declare no conflict of interest.

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
