# Peer review of "Recent Advances in the Hydroxylation of Amino Acids and Its Derivatives"

_fermentation, doi:10.3390/fermentation9030285_

Round 1

Reviewer 1 Report

This review manuscript summarizes the biocatalytic synthesis of hydroxy amino acids and their derivatives. There are a few minor suggestions:

1. It would be helpful for the readers to understand the variety of hydroxy amino acids if the authors could add a figure to show the chemical structures of hydroxy amino acids and their derivatives that are described in this manuscript.

2. The authors use Tyr as an abbreviation for tyrosol. However, Tyr generally stands for tyrosine. Therefore, it is better to use a different abbreviation for tyrosol.

3. There are some typos throughout the manuscript. For example, "-KG" should be "α-KG" (lines 187-188). "-lactamase" should be "β-lactamase" (line 168).

4. line 64-66: "The naturally occurring L-2-hydroxy-phenylalanine and L-3-hydroxy-phenylalanine are rare, and could be generated by the non-enzymatic free radical hydroxylation of phenylalanine under oxidative stress." The authors could describe the enzymatic reaction of phenylalanine 3-hydroxylase to synthesize L-3-hydroxy-phenylalanine (doi: 10.1021/bi200733c).

Author Response

Reviewer #1

This review manuscript summarizes the biocatalytic synthesis of hydroxy amino acids and their derivatives. There are a few minor suggestions:

Comment 1: It would be helpful for the readers to understand the variety of hydroxy amino acids if the authors could add a figure to show the chemical structures of hydroxy amino acids and their derivatives that are described in this manuscript. 

Response: Thank you for your comments and suggestions. In page 3, we have added a column in Table 1 to show the chemical structures of hydroxy amino acids and their derivatives that are described in this manuscript in Table 1.

Comment 2: The authors use Tyr as an abbreviation for tyrosol. However, Tyr generally stands for tyrosine. Therefore, it is better to use a different abbreviation for tyrosol.

Response: Thank you for your comments and suggestions. Sorry, we made such a mistake. Now, the full names of tyrosol and hydroxytyrosol are used in the revised manuscript. On page 2, 3 and 9, “HTyr” was revised as “hydroxytyrosol” and “Tyr” was revised as “tyrosol”.

Comment 3: There are some typos throughout the manuscript. For example, "-KG" should be "α-KG" (lines 187-188). "-lactamase" should be "β-lactamase" (line 168).

Response: Thank you for your comments and suggestions. Sorry, we made such a mistake. In line 168 on page 3, "-lactamase" was revised as “β-lactamase”. In lines 187 and 188 on page 3, "-KG" was revised as "α-KG"

Comment 4: line 64-66: "The naturally occurring L-2-hydroxy-phenylalanine and L-3-hydroxy-phenylalanine are rare, and could be generated by the non-enzymatic free radical hydroxylation of phenylalanine under oxidative stress." The authors could describe the enzymatic reaction of phenylalanine 3-hydroxylase to synthesize L-3-hydroxy-phenylalanine (doi: 10.1021/bi200733c)

Response: Thank you for your comments and suggestions.

In lines 2-7 on page 2, “Of course, enzymatic catalysis has also been reported for the synthesis of L-3-hydroxy-phenylalanine. For example, the Fe2+-dependent phenylalanine hydroxylase (PheH) can catalyze the hydroxylation of aromatic amino acid L-phenylalanine to L-3-hydroxy-phenylalanine” has been add after "The naturally occurring L-2-hydroxy-phenylalanine and L-3-hydroxy-phenylalanine are rare, and could be generated by the non-enzymatic free radical hydroxylation of phenylalanine under oxidative stress."

Reviewer 2 Report

The review article provided the biosynthetic hydroxylation of aliphatic, heterocyclic, and aromatic amino acids and introduced the basic research and application of hydroxy amino acids. I find the middle of the paper quite good, but the abstract, and introduction, are quite poor currently and do little to provide adequate background, context, and clarity for why the authors chose to review the hydroxylation of amino acids and its derivatives. The manuscript should provide the conclusion and further perspective on the hydroxylation of amino acids.

The listed are some comments regarding the submitted manuscript:

1.      Line 22-41: The introduction part should be rewritten. This part should provide the bioactivity of the hydroxylation of amino acids.

2.      Line 374, Table 1: Since Corynebacterium glutamicum is well known for its extraordinary l-amino acid production properties, why E. coli was choosing to produce amino acids and their derivatives in these data?

Author Response

Reviewer #2

The review article provided the biosynthetic hydroxylation of aliphatic, heterocyclic, and aromatic amino acids and introduced the basic research and application of hydroxy amino acids. I find the middle of the paper quite good, but the abstract, and introduction, are quite poor currently and do little to provide adequate background, context, and clarity for why the authors chose to review the hydroxylation of amino acids and its derivatives. The manuscript should provide the conclusion and further perspective on the hydroxylation of amino acids.

Comment 1:  The review article provided the biosynthetic hydroxylation of aliphatic, heterocyclic, and aromatic amino acids and introduced the basic research and application of hydroxy amino acids. I find the middle of the paper quite good, but the abstract, and introduction, are quite poor currently and do little to provide adequate background, context, and clarity for why the authors chose to review the hydroxylation of amino acids and its derivatives. The manuscript should provide the conclusion and further perspective on the hydroxylation of amino acids.

Response: Thank you for your comments and suggestions.

In line 360 on page 8, we have added “a conclusion” as follows:

HAAs are an important class of functional amino acids with a variety of functions and are widely used in feed, pharmaceutical, building and chemical industry. It not only has antifungal, antibacterial, antiviral and anticancer effects, but also can be used as a precursor or intermediate in the asymmetric synthesis of drugs, and can also be used as a component of chiral compounds. Based on the widespread use of HAAs, the development of engineered enzymes to control the selective synthesis of HAAs is one of the indispensable methods to promote the yield of HAAs. Many engineered enzymes have been found to hydroxylate free amino acids into corresponding HAAs. However, the biocatalytic synthesis of HAAs is still limited to a single enzyme and a single carrier. Therefore, new members of functional amino acid hydroxylases should be characterized by methods involving gene mining, bioinformatics and high-throughput screening techniques.

In lines 24-28 and 32-34 on page 1, we have added “Amino acid (AA) is a cell signal molecule with significant metabolic and regulatory functions. They are necessary precursors for the synthesis of various important molecules. They can be converted into various derivatives through halogenation, N-alkylation, hydroxylation and other ways. These derivatives have important roles in various fields such as medicine and chemical industry. Among them, hydroxy amino acids play an important role.” in front of the introduction and “In this context, developing engineered enzymes to control the selective synthesis of HAAs becomes an attractive pathway for amino acid hydroxylation.” was revised as “In this context, the development of engineering enzymes to control the selective synthesis of HAAs has become an attractive way for amino acid hydroxylation.”

In lines 40-41 on page 1, we have added “hydroxyl-lysine, trans-4-hydroxyl-proline and hydroxyl-isoleucine are good pharmaceutical raw materials” in front of “L-5-hydroxytryptophan (5-HTP) increases melatonin levels and promotes sleep”.

In lines 14-22 on page 1, we have modified the abstract.

Comment 2:   Line 374, Table 1: Since Corynebacterium glutamicum is well known for its extraordinary l-amino acid production properties, why E. coli was choosing to produce amino acids and their derivatives in these data?

Response: Thank you for your comments and suggestions.

In Line 374, Table 1, we have added some hydroxyl amino acids and their derivatives produced by Corynebacterium glutamicum in Table 1.

Round 2

Reviewer 2 Report

I have agreed the revised manuscript for publication